# Cancer Loyalty Card Study (CLOCS): protocol for an observational case–control study focusing on the patient interval in ovarian cancer diagnosis

Hannah R Brewer,[1] Yasemin Hirst,[2] Sudha Sundar,[3,4] Marc Chadeau-Hyam,[5,6] James M Flanagan  [1]

¹Surgery and Cancer, Imperial College London, London, UK
²Behavioural Science and Health, University College London, London, UK
³Institute of Cancer and Genomic Sciences, University of Birmingham, Birmingham, UK
⁴Pan-Birmingham Gynaecological Cancer Centre, City Hospital, Birmingham, UK
⁵MRC Centre for Environment and Health, Imperial College London, London, UK
⁶Epidemiology and Biostatistics, School of Public Health, Imperial College London, London, UK

**Correspondence to**
Dr James M Flanagan;
j.flanagan@imperial.ac.uk

## ABSTRACT

**Introduction** Ovarian cancer is the eighth most common cancer in women worldwide, and about 1 in 5 women with ovarian cancer do not receive treatment, because they are too unwell by the time they are diagnosed. Symptoms of ovarian cancer are non-specific or can be associated with other common conditions, and women experiencing these symptoms have been shown to self-manage them using over-the-counter medication. Results from a recent proof-of-concept study suggest there may be an increase in the purchases of painkillers and indigestion medication 10–12 months before ovarian cancer diagnosis. We propose a case–control study, as part of a larger project called the Cancer Loyalty Card Study (CLOCS), to investigate whether a significant change in medication purchases could be an indication for early signs of ovarian cancer, using data already collected through store loyalty cards.

**Methods and analysis** Using a retrospective case–control design, we aim to recruit 500 women diagnosed with ovarian cancer (cases) and 500 women without ovarian cancer (controls) in the UK who hold a loyalty card with at least one participating high street retailer. We will use pre-existing loyalty card data to compare past purchase patterns of cases with those of controls. In order to assess ovarian cancer risk in participants and their purchase patterns, we will collect information from participants on ovarian cancer risk factors and clinical data including symptoms experienced before diagnosis from recruited women with ovarian cancer.

**Ethics and dissemination** CLOCS was reviewed and approved by the North West-Greater Manchester South Research Ethics Committee (19/NW/0427). Study outcomes will be disseminated through academic publications, the study website, social media and a report to the research sites that support the study once results are published.

**Trial registration number** ISRCTN 14897082, CPMS 43323, NCT03994653.

### Strengths and limitations of this study

► This study is a novel approach to investigate whether routinely collected commercial purchase data can be used to reduce delays in ovarian cancer diagnosis.
► Ovarian cancer advocates have been involved in the conception, design and conduct of the study.
► A key strength of the study is the detailed information available for purchases recorded for up to 6 years prior to enrolment.
► A key limitation is that loyalty card holders may not use their cards for every purchase, may shop at other stores not being accessed in this study and may be purchasing items for other family members.
► In general, due to the retrospective nature of case–control studies, there is potential for the misreporting of exposures in the risk factor questionnaire by cases and controls (recall bias).

primary and secondary care settings among symptomatic patients is a challenge due to their low positive predictive value of symptoms and symptom presentation at various stages of the ovarian cancer prognosis.[2 3] Symptoms such as increased abdominal size, pelvic pain, abdominal pain and bloating, feeling full quickly and difficulty eating have been identified as high-risk symptoms but equally it has been reported that the absence of a symptom does not help in ruling out ovarian cancer.[4 5] For patients whose symptoms do not warrant an immediate referral earlier presentation in primary care is crucial to reduce delays in diagnosis and subsequent treatment.

However, symptom recognition and help seeking in primary care are also not without challenges. While in some cases patients may present very quickly, for some patients it could take up to 12 months before they seek any medical help.[6] It has been shown that greater awareness of symptoms alone does not facilitate help seeking for ovarian

## INTRODUCTION

Approximately 7200 new cases of ovarian cancer are diagnosed each year in the UK, and with over 4200 women dying from the disease each year, it is a particularly lethal form of cancer.[1] Diagnosing ovarian cancer in

cancer because symptoms are often perceived as vague or trivial.[7 8] The major reasons for not presenting to the primary care are often highlighted as 'not wanting to waste the GP's time',[9] normalisation of these symptoms[10] and few women believing that their symptom(s) might be a sign of cancer.[11] Women might choose to self-manage such symptoms by seeking advice from sources other than a primary care physician, for example, pharmacists, family, internet and using over-the-counter medication,[12] which may also have been moderated by other factors such as access to healthcare, sociodemographic factors and other competing priorities, for example, work and care responsibilities.[7–11]

There are currently no organised screening programmes offered to women that could help detect ovarian cancer at an earlier stage, and there is little evidence for effective interventions aiming to improve symptom recognition and help seeking.

Therefore, novel methods are needed to improve earlier detection of low-volume disease or earlier stage to improve survival.[13] A recent proof-of-concept study investigated whether purchase information collected through the high street retailers' loyalty cards (eg, store point cards) could be used to identify women's self-care behaviours prior to presenting to healthcare professionals. This study included data from a major high street retailer's loyalty card of women who have previously been diagnosed with ovarian cancer and consented their past purchase data to be provided to the researchers. The study suggested that there may be an increase in the purchases of painkillers and indigestion tablets up to 10–12 months before diagnosis.[14] This proof-of-concept study provided a new model of investigation for symptom recognition and help seeking among patients with ovarian cancer. It highlighted potential strengths and limitations in its methodology for future research. Building on these results, self-care behaviours of patients with ovarian cancer will be further investigated with a case–control study within a larger project called Cancer Loyalty Card Study (CLOCS). CLOCS hypothesises that a significant change in purchasing behaviour, primarily an increase in self-medication or self-care behaviours to treat symptoms prior to presentation in primary care (see figure 1), could be an indication for early signs of ovarian cancer.

## Study aims

The primary aim of this case–control study is to investigate prediagnosis purchasing behaviours of patients with ovarian cancer (cases) with women without a previous ovarian cancer diagnosis (controls) using data collected through two major high street retailers' loyalty card schemes. We will investigate purchase behaviour of items that are suspected to treat symptoms of ovarian cancer (eg, painkillers, indigestion tablets) as well as a change in frequency or quantity of other purchased items. The secondary aim of CLOCS will be to investigate whether or not a purchase threshold as an 'alert' about potential ovarian cancer symptoms in individuals can be possible using loyalty card data and determining the predictive utility of purchasing behaviours in the early detection of ovarian cancer based on ovarian cancer risk. This would be important to inform if future interventions would be possible using loyalty card data. The findings from this retrospective study will need to be validated in a prospective cohort study prior to any clinical impact being realised. However, the clinical impact of this study for ovarian cancer diagnosis may involve a number of potential interventions. Future interventions could include an alert sent to individuals who exceed the 'alert threshold' to visit a general practitioner (GP) to discuss their symptoms, or a potential public health message about those purchase thresholds and self-treatment of ovarian cancer symptoms.

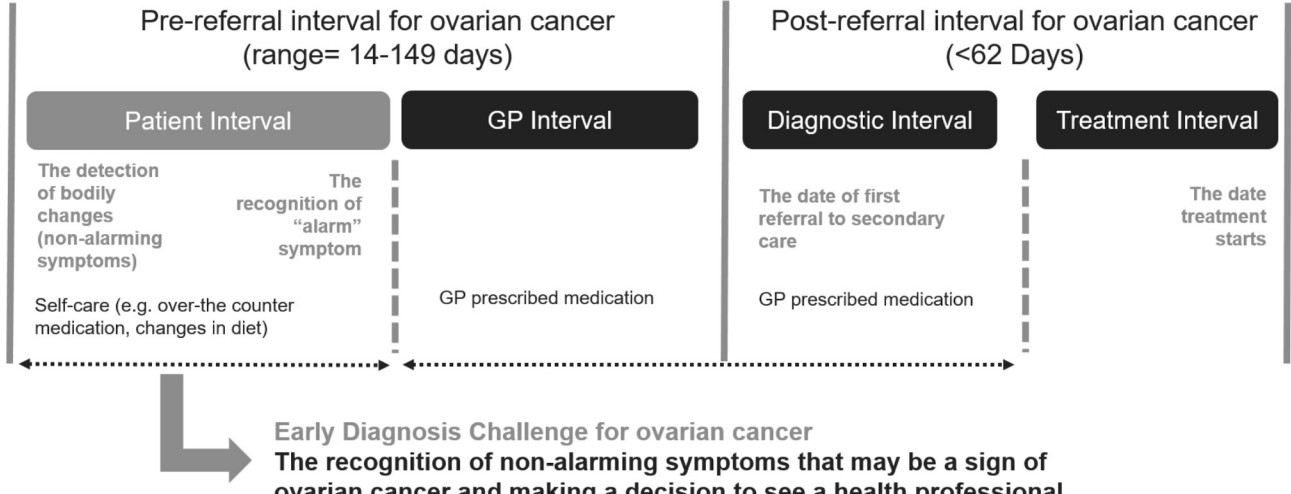

**Figure 1** Pathway to diagnosis highlighting the patient interval that Cancer Loyalty Card Study (CLOCS) will investigate. GP, general practitioner.

## METHODS AND ANALYSIS
### Study design and setting
CLOCS will use a retrospective case–control design to compare past purchase patterns of women with and without ovarian cancer using already existing data gathered by two major high street retailers, one of which is a health and beauty retailer and the other a groceries and general merchandise retailer. Furthermore, we will collect data on risk factors for ovarian cancer, symptoms experienced before diagnosis and clinical data from patients with ovarian cancer using self-reported questionnaires.

CLOCS will be recruiting across the UK from at least 40 National Health Service (NHS) trusts and hospitals and recruit healthy volunteers using online recruitment.

### Study timeline
The CLOCS case–control study will take place from 1 November 2019 until 1 February 2022.

### Participants
#### Inclusion/exclusion criteria
##### Cases
Women, at least 18 years old, diagnosed and living with any type of ovarian cancer (including all subtypes of epithelial ovarian cancer high-grade serous, low-grade serous, endometrioid, clear cell, mucinous, borderline and other subtypes) within the previous 2 years from the recruitment date who own at least one participating high street retailer loyalty card are eligible to join CLOCS case–control study as cases.

There are no further clinical criteria for patients to be excluded from this study. Patients' usual care will continue to be provided by the healthcare professionals, and CLOCS participation does not exclude patients from participating in other clinical or non-clinical studies.

##### Controls
Women, at least 18 years old, who have not been diagnosed with ovarian cancer and own at least one of the loyalty cards from the participating retailers are eligible to join the CLOCS as controls. We are not excluding women with previous cancer diagnosis, or women who may have had their ovaries removed for reasons other than cancer; however, these will be considered when cases are matched with controls.

#### Recruitment
The CLOCS case–control study recruitment process is outlined in figure 2 and described in further detail below.

##### Cases
Eligible patients with ovarian cancer will be recruited by members of their healthcare teams at approximately 40 NHS trusts and hospitals across the UK that have agreed to approach patients at clinics. These healthcare teams are registered as research sites and will have a principal investigator (PI) leading the recruitment to CLOCS. The PI and the research nurses will facilitate recruitment by identifying patients when they think it is suitable to approach patients based on patients' overall well-being at a suitable time (eg, at waiting areas, during chemotherapy sessions), then will provide patients with the study information to consider participation. Patients will be given a paper-based participant pack that includes study consent form including consents for providing access to data collected by the retailers, self-report questionnaire and a clinical questionnaire to be completed by the clinical staff facilitating the recruitment. If they wish to participate, patients can complete all the study documents at the recruitment site and post it back to the research team using the free-post envelope provided in the study pack, or they are also given the option to consider participation at home provided that the clinical questionnaire has been completed by a member of their clinical team. The research sites will not need to record any data from the participants. The data collection from the cases will be a one-off process and no further contact with the patient is required for the study.

##### Controls
The control participants to the study will be recruited through the study website (www.clocsproject.org.uk) where data collection will be safeguarded using an encrypted web form that includes study consent and the self-report questionnaire. The website has been developed and maintained by NDP. The questionnaire for women without ovarian cancer will be confined to risk factors. The online participation will take up to 15 min and data will be transferred immediately to the secure CLOCS server (see figure 3). The website approach was adopted as one of the outcomes of the proof-of-concept study. It aims to provide a transparent environment where potential participants can make informed decisions about participation in CLOCS, and the data risks are minimised using encryption. We have carefully designed the website to inform the general public about how we will process their data in CLOCS, how they can withdraw from the study if they wish to do so and use it as a platform to provide feedback as we progress in the larger CLOCS project. The research team will use several methods to recruit control participants to the study which will include a press release carried out by the project funders and the Imperial College London public engagement team, dissemination of study flyers and leaflets to patients in participating NHS recruitment clinics and social media marketing techniques on Twitter and Facebook using paid advertisement and unpaid methods. The application of the social media recruitment methods will allow researchers to reach a larger cohort in the UK. The application of both paper-based and online recruitment methods will be carried out throughout CLOCS and will be recorded. The efficacy of these methods will be reported in peer-reviewed journals to inform online research recruitment to future case–control studies.

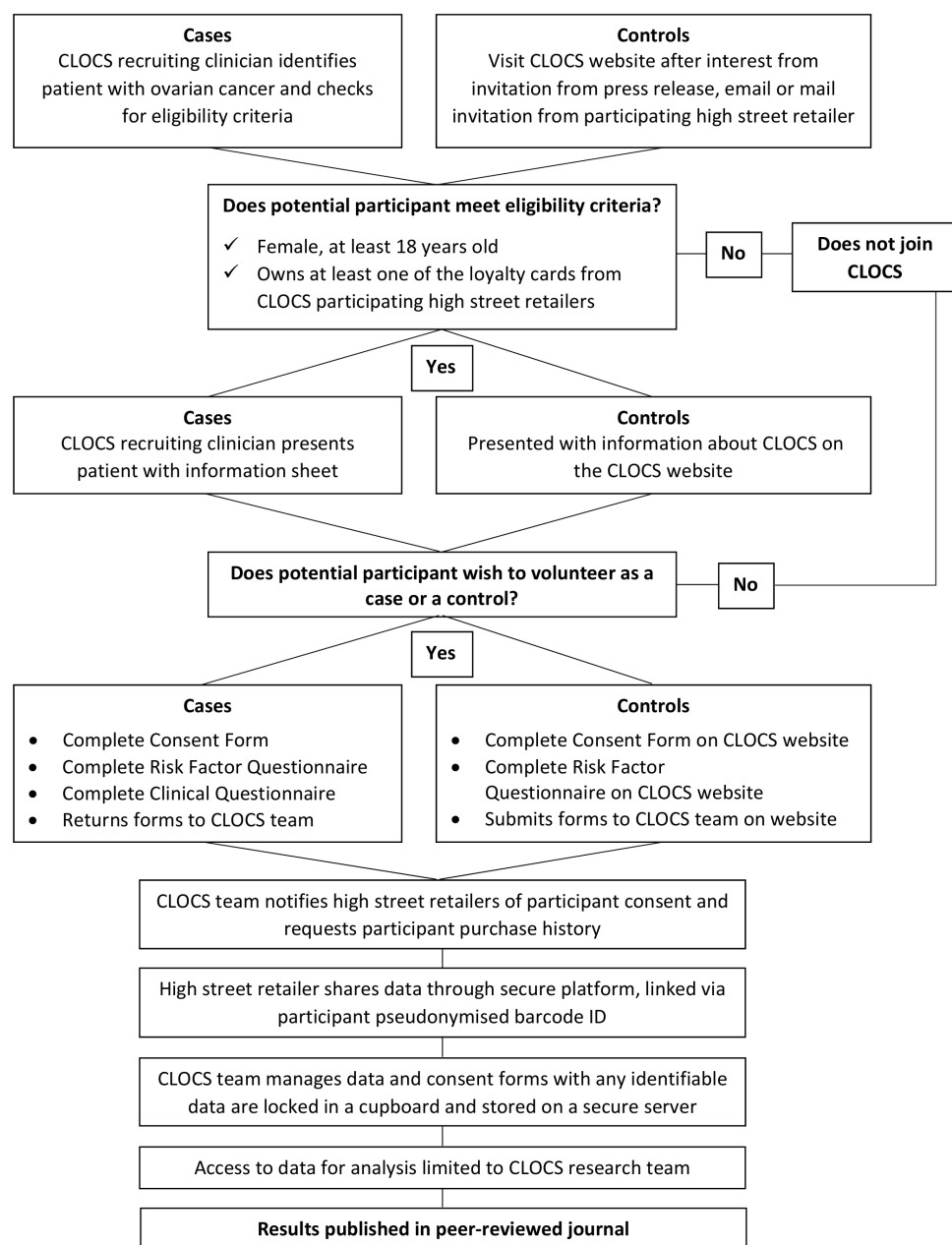

**Figure 2** Cancer Loyalty Card Study (CLOCS) recruitment process.

## Data sources and variables included
### Loyalty card data (for cases and controls)

CLOCS participants will hold a loyalty card at one or both participating retailers, including two health, beauty and/or grocery retailers that can be accessed by the UK population within all precincts allowing the study to assess purchases of painkillers and indigestion medication as well as eating habits. The study will not include prescription-based pharmacy data. Once participants consent for the CLOCS team to request their loyalty card data from the retailers on their behalf, the study will include individuals' past purchase data from up to 6 years prior to the date of study consent. The purchase data will include each individual purchase, date of purchase, the location (ie, store postcode) and the product categorisation that is provided by the retailers. There will be no additional individual information recorded by the study team from this data set and no additional individual information will be requested from the retailers. Please note that at this stage of the study, we will not be naming the high street retailers in publications and in our direct communications, but they are named to the potential participants when they agree to take part.

### Ovarian cancer risk questionnaire (for cases and controls)

In order to stratify the purchase differences between cases and controls based on ovarian cancer risk, we designed a questionnaire on ovarian cancer risk factors that will allow us to determine individuals' ovarian cancer risk. Both cases and control participants will be asked to complete a short questionnaire about ethnicity, marital status and the following well-established ovarian cancer

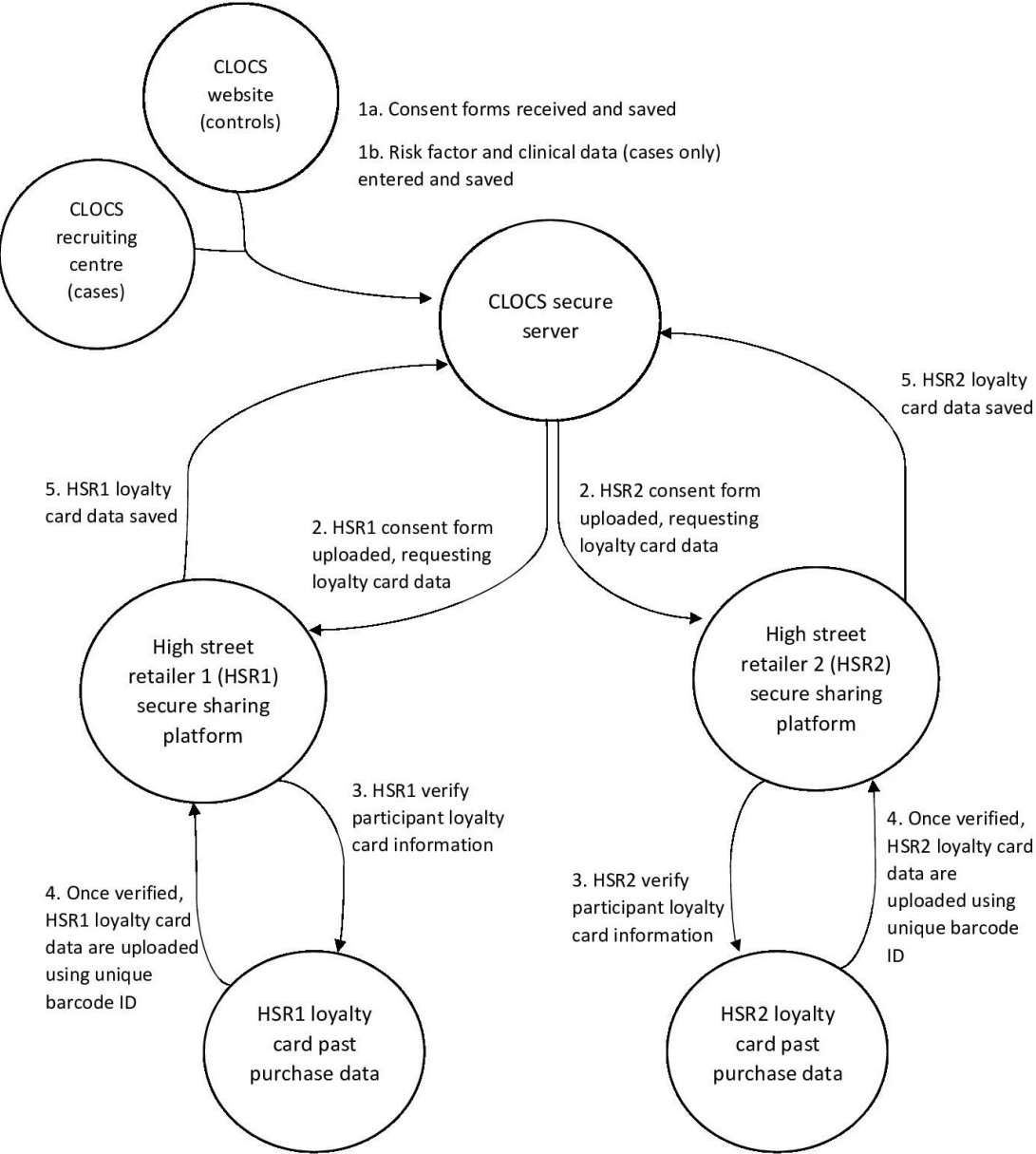

**Figure 3** Cancer Loyalty Card Study (CLOCS) data flow chart.

risk factors: body mass index, age at menarche, menopausal status, age at menopause, parity, breast feeding, hysterectomy, tubal ligation, cancer history, endometriosis, aspirin use, oral contraceptive use, hormone replacement therapy (HRT), family history of ovarian and breast cancers, vaping and cigarette smoking.[15] These factors will be used to assess ovarian cancer risk among participants to distinguish high-risk and low-risk individuals and subsequently observe patterns in purchase history stratified by risk levels.

The questionnaire also includes questions about the symptoms experienced (if any) and number of visits to the GP in the year leading up to cancer referral or diagnosis which were adopted from the proof-of-concept study.[14]

The final section of the questionnaire contains questions about which food and pharmacy stores participants

shop at regularly, at which stores they own a loyalty card and frequency of use of those loyalty cards.

### Clinical questionnaire (cases only)

Women with an ovarian cancer diagnosis will also have a clinical questionnaire completed by their healthcare team. This will provide the objective timeline and pathway for diagnosis (eg, GP referral, emergency presentation, other), details on the type (high-grade serous, low-grade serous, endometrioid, clear cell, mucinous, borderline, other), stage (I, II, III, IV), grade of ovarian cancer at diagnosis (grade 1 or well differentiated, grade 2 or moderately differentiated, grade 3 or poorly differentiated/undifferentiated), surgical outcome if any (optimal debulking, suboptimal debulking, no surgery) and BReast CAncer susceptibility gene (BRCA) status if the participant has been tested (BRCA1 positive, BRCA2 positive,

both negative, awaiting result). We will use this information in risk assessment and to stratify the loyalty card data of cases and controls based on the time of diagnosis, and observe purchase history stratified by type, stage and grade of ovarian cancer diagnoses.

### Data collection processes

In the next section, we will describe how we will be collecting and processing data in the CLOCS case–control study (figure 3).

### Data acquisition

On choosing to participate, both cases and controls will need to complete a consent form to participate in CLOCS and return it to the CLOCS research team either by post (cases only) or the CLOCS website (controls only). The CLOCS consent form includes two sections. The first section details consent to take part in CLOCS, and the second section is for participants to consent to the CLOCS research team to request their data from high street retailers on their behalf. The consents for loyalty cards include the exact name of the individual and the exact membership number as written on their card. All individual consents including scanned copies of paper consents will be saved in the secure and encrypted server based in the 'Secure Enclave' at Imperial College London. Participant loyalty card details will be entered into a comma-separated values (CSV) file and encrypted for secure data request from retailers using a secure file transfer protocol (SFTP).

### Data extraction from the high street retailers

The CLOCS team will send the CSV file that contains a unique study ID, name of the individual and the exact loyalty card number specific to the retailer using the secure file transfer. The respective high street retailer will validate the participant information from the consent form with their loyalty card data. If loyalty card details cannot be validated, the consents and any information from those individuals who have not consented to be recontacted will be excluded from the study and deleted immediately. High street retailers will then generate a data set for all consenting individuals and will share the past purchase data with the CLOCS team using the SFTP. Depending on the high street retailer's records and when the participant obtained the loyalty card, up to 6 years of participants' past purchasing data prior to the date of recruitment will be shared.

### Data merge

The CLOCS team will receive the encrypted participant purchase history from the high street retailers through the SFTP, save and store all data to a secure and encrypted server based in the Secure Enclave at Imperial College London, described in the Data Management section. All data will be pseudonymised and analysed on the secure server.

### Data deletion

The information relating to the participants' consents for providing past purchase data will be retained in line with each high street retailer's data retention policy. Ten years after the study is complete, all paper-based documents will be shredded and destroyed in accordance with Imperial College London's data deletion and retention processes. Electronically held identifiable data including individual consent and contact details will be kept for 10 years after the study is complete and will be deleted from the secure server in accordance with Imperial College London's data deletion and retention processes.

Past purchase data will be kept until the results are published or 5 years from the date of participation in the study, whichever comes sooner, before it will be anonymised and the link to all identifiable data will be deleted. During the lifetime of the study, participants are given the option to withdraw consent which will mean that all data associated with that individual will be deleted and records will be removed from CLOCS.

### Sample size and statistical power

Based on the modelled data from the pilot study,[14] we hypothesise that we could detect a difference in purchase proportions of 8.3% at 3 months prior to diagnosis, 6.3% at 6 months, 4.3% at 9 months and 2.3% at 12 months prior to diagnosis between cases and controls. Using the two-sample test for proportions the minimum group size to detect these differences at 80% power and $\alpha=0.05$ would be 125 (−3 months), 185 (−6 months), 328 (−9 months) and 969 (−12 months). We modelled the power versus time prior to diagnosis and chose 10 months as our target to retain sufficient power for larger effect sizes at earlier time points.

Therefore, in the final analysis we will have >80% power at $\alpha=0.05$ to detect an increase in proportion of purchases in cases (>4.7%) versus controls (1.4%) at 10 months prior to diagnosis, based on the modelled data, with a minimum n=431 cases. We aim to recruit 500 participants to account for 10%-15% of very low-frequency card users. We will compare proportions using Fisher's exact test comparing target (items suspected to be used to treat ovarian cancer symptoms) and non-target (all other items) purchases in cases at each time point prior to diagnosis compared with the average monthly proportions in the controls. Target items for analysis will include but will not be limited to medications for pain, indigestion and bloating relief (see online supplementary appendix A table 1 for a list of example purchases). We will report proportions, 95% CIs and p values.

### Statistical methods

We will use descriptive statistics to provide details on the characteristics of the cases and controls by sociodemographic (eg, age, ethnicity) and by risk factors (eg, parity, oral contraceptive use, and so on). We will report feasibility factors such as number of people who had both, or

one loyalty card, and number of years of data included in the study.

Cases and controls will be matched 1:1 by age (5-year age groups) and ethnicity. Ethnicity will be matched based on the guidance from the Office for National Statistics.[16] We will aim to define the time by which cases and controls become statistically significantly different in their purchase behaviours using a multivariable, conditional logistic regression model with ovarian cancer as the outcome and 'purchase proportion' at different time points to diagnosis as the exposure, adjusting for stage and histology and other potentially confounding variables collected from the risk factor questionnaire (including body mass index, age at menarche, menopausal status, age at menopause, parity, breast feeding, hysterectomy, tubal ligation, cancer history, endometriosis, aspirin use, oral contraceptive use, HRT, family history of ovarian and breast cancers, vaping and cigarette smoking[15]). We will compare the proportions of purchases in cases to controls, stratified by risk quintiles defined by the risk model (estimated n=86 per quintile) and by histology (with expected proportions of high-grade serous, mucinous cases, endometrioid, clear cell and low-grade serous cases). Analyses will be performed for different types of medications separately and combined (eg, a comparison of case–control purchase proportion of painkillers alone and indigestion medications alone, and then of painkillers and indigestion medications combined).

Established risk factors have been reported to modestly discriminate between cases and controls (area under the curve=0.66),[17] and polygenic risk scores only slightly improve these performances. This model has however been proposed to improve epithelial ovarian cancer detection with risk-stratified screening.[18] We will establish the risk profile, a mathematical summation of all of the risk factors, of each participant based on the published method.[17] Our study will be underpowered to establish a difference between high and low-risk groups (20% of subjects per quintile of risk), it is therefore exploratory. We will estimate the predictive values of this 'early diagnosis' test for each stratum. If this analysis indicates a potential difference, then that will guide future studies that will be sufficiently powered to address this.

We will conduct exploratory analyses investigating other methods leveraging the autocorrelation in the longitudinal data. Specifically, we will explore the evolutions of individual frequency of purchases and purchase trajectories, to identify purchase patterns (eg, increasing purchases before diagnosis, purchase peaks, and so on), that are indicative/predictive of the disease onset and of the time to onset.

All analyses will be done using the statistical software R.

### Interim analysis

An interim analysis will be conducted after 6 months of CLOCS recruitment to provide an initial exploratory analysis of the data and assess the demographics of CLOCS participants. This will help determine whether recruitment will need to target specific demographics (eg, age groups) in order to ensure there is a representative distribution of ages and ethnicities for the matched analysis.

### Study risks and sources of bias

There are no clear risks for participants in the CLOCS. However, it may be emotional or distressing for patients to think about cancer and the first time they noticed some changes in their body (ie, symptoms). The CLOCS team intended to design the survey questions in the most sensible and sensitive way to ensure that there are no negative effects of this study on patients' well-being. All participants are advised to see a medical professional in the information sheet if they are worried about a symptom or sign that they might have been experiencing.

For many loyalty card holders, purchases will be for themselves and other individuals in their household. The CLOCS team will not be able to quantify what purchases are consumed by the participant and by other members of the household, but the team will be able to make assumptions to adjust accordingly.

The CLOCS team acknowledges that not all participants will have used their loyalty card with every purchase for various reasons (eg, card not with them at time of purchase, in a hurry, and so on) and that participants will have made purchases at other shops not recorded with their loyalty cards observed in this study. For this reason, the risk factor questionnaire asks participants to indicate how often they use their loyalty card when shopping, at which stores they shop most regularly and own a loyalty card, so that these can be taken account at the analysis stage.

It is possible that CLOCS participants could have other comorbidities, and their over-the-counter purchases might be used to treat symptoms from those instead of symptoms of ovarian cancer. If women have been diagnosed with comorbidities (eg, asthma, diabetes, osteoporosis, depression, and so on[19]), we would assume that they might already be receiving prescribed medication to treat these from their healthcare providers. If they are using prescribed medication, then loyalty card data do not include information about prescriptions. Common comorbidities that are not yet diagnosed or that do not require prescription medication are expected to be present in similar frequencies among CLOCS participants with and without ovarian cancer. The risk factor questionnaire will address some of these comorbidities, which will allow us to adjust for them in the analysis.

It is also possible that patients with ovarian cancer may have already been seen by a healthcare professional for their symptoms and may have been prescribed medication for reasons other than ovarian cancer, for example, urinary tract infection, irritable bowel syndrome, prior to being referred for suspected ovarian cancer. However, access to prescribed medication information would require access to primary care records and not included in this study. It would also mean that patients have already

sought help for their symptoms and the patient interval phase for symptom appraisal before help seeking would have been finalised. We will control for this by adjusting for the self-reported number of visits to the GP leading up to the diagnosis. This would also be discussed as a potential limitation, and we will be cautious in our evaluations of the results if we observe a reduction in risk-related purchases close to the date of diagnosis.

CLOCS limits participation to women who own a loyalty card from at least one of the participating retailers. Approximately 73% of women aged 50–80 years in the UK have a loyalty card to one of the participating high street retailers, and therefore the cardholders in CLOCS should be representative. Nevertheless, there will be selection bias against those who do not hold a loyalty card.

In any retrospective case–control study, there is the potential for recall bias. Participants with ovarian cancer could report experiencing more symptoms more frequently than may have occurred, but for those who report they self-managed these symptoms, the CLOCS team will be able to observe loyalty card purchase data to validate the amount and frequency of self-medication prior to diagnosis.

### Data management

The CLOCS research team will act as *data processors* when requesting data from high street retailers and the high street retailers would be *data controllers*. CLOCS will become the data controller for the self-reported questionnaire data and the clinical questionnaire data. The consenting participants will be *data owners* and will have the right to withdraw their consent from the study at any time without giving any reason to the study team.

Only the CLOCS research team will have access to the paper-based consent forms which will be kept in a locked cupboard and server in a locked room and secure building that requires security badge access to enter. The data will be analysed in a secure environment with access limited to the research team. Participants will be given a unique study ID bar code which means all data sets will be pseudonymised.

All data will be stored and processed in Imperial College's ISO27001 certified secure environment—the 'Secure Enclave'. This is a fully managed infrastructure and secure environment providing high availability, resilience and business continuity through multiple servers, backups and disaster recovery measures. A robust data security model has been designed to protect sensitive personal and medical data from the potential risk of unauthorised access or distribution. No one other than the research team will have access to the research data set.

The CLOCS website is compliant with the high level of data security required by Imperial College London and General Data Protection Regulation. The data collected through the website only reside in the Secure Enclave and not in the browser or website. It also uses public key encryption, which encrypts data from the site that can only be unlocked once within the enclave. These

requirements are equalled by support for the user to complete the online survey, by using content and interfaces that are easy to use, especially on mobile phones.

We have strict data processing agreements in place with our commercial partners to ensure that data are protected at all times. Commercial partners will not have access to the study data and will not financially benefit from this project.

### Patient and public involvement

Patients with ovarian cancer and members of the public have been involved in the development of the CLOCS from the beginning, participating in the proof-of-concept study that assessed the general public's acceptability of the use of commercial and health data linkages for cancer symptom surveillance.[14] Two patient advocates with ovarian cancer who participated in the pilot study have taken part in research meetings developing this case–control study and reviewed study documentation. They have been and will continue to be involved in ongoing development of CLOCS methods and dissemination of the research results through social media, media outlets and patient groups. The questionnaire for women without ovarian cancer has also been reviewed and approved by members of the public.

We also use our website for providing information to the patients and public. For instance, all research goals, how we will be analysing data, confidentiality and data security are all detailed for the lay population. There is an animation hosted on our website that is aiming to explain the study in simpler words. All materials are reviewed and approved by patient advocates. We carry out annual research meetings with stakeholders including research nurses, patients, academic collaborators, retailers, funders and public to provide updates and receive feedback on this project.

### ETHICS AND DISSEMINATION

CLOCS was reviewed and approved by the North West-Greater Manchester South Research Ethics Committee (19/NW/0427). The study is registered as an NHIR portfolio study, with the ISRCTN, and is registered on ClinicalTrials.gov. This paper describes the ethics approved protocol dated 8 August 2019 (version 2).

We will disseminate the outcomes of the study using academic publications, the study website (clocsproject.org.uk), social media (@CLOCS_Imperial), and send a report to the research sites that supported the study once the results are published. We will aim to publish the results in a peer-reviewed journal in accordance with the Strengthening the Reporting of Observational Studies in Epidemiology (STROBE) statement for case–control studies.[20] The STROBE statement we intend to use when reporting the results from this case–control study is listed in online supplementary appendix B. We will also use the study website to inform the public about the study outcomes.

## Data sharing policy

All access to the CLOCS data collected from the retailers will be restricted and will not be shared with other researchers due to the data sharing agreements with the retailers. However, we do encourage other researchers to contact us for collaborations. If they are interested, they are given the opportunity to review the metadata by directly contacting us and submit a research proposal where the data analyses will be carried out by a member of the CLOCS team. All proposals will be reviewed case by case by the PI and the rest of the research team.

**Acknowledgements** We thank NDP Studio for their support and development of the study website (www.clocsproject.org.uk). We thank our patient and public representatives for reviewing the study materials and providing guidance on study acceptability from the conception of CLOCS to data collection. The authors acknowledge the infrastructure and recruitment support from the Imperial Experimental Cancer Medicine Centre, Cancer Research UK Imperial Centre, the National Institute for Health Research Imperial Biomedical Research Centre and the Ovarian Cancer Action Research Centre. We thank Boots UK employees who facilitated the project for their support and guidance during the development of this project. We thank Eric Johnson for guidance on data security and GDPR.

**Contributors** JMF and YH conceived the study. HRB wrote the first draft of the manuscript. YH, JMF, SS and MCH commented and reviewed the final draft of the manuscript. All authors agree on the final version of the protocol.

**Funding** Cancer Loyalty Card Study (CLOCS) is funded by Cancer Research UK (CRUK) Early Detection Project Grant (C38463/A26726). YH and HRB are funded by CRUK.

**Competing interests** None declared.

**Patient and public involvement** Patients and/or the public were involved in the design, or conduct, or reporting, or dissemination plans of this research. Refer to the Methods section for further details.

**Patient consent for publication** Not required.

**Provenance and peer review** Not commissioned; externally peer reviewed.

**ORCID iD**
James M Flanagan http://orcid.org/0000-0003-4955-1383

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
