## [Reviewer comments · BMJ Open]

ARTICLE DETAILS

TITLE (PROVISIONAL)	Cancer Loyalty Card Study (CLOCS): protocol for an observational case-control study focusing on the patient interval in ovarian cancer diagnosis
AUTHORS	Brewer, Hannah; Hirst, Yasemin; Sundar, Sudha; Chadeau-Hyam, Marc; Flanagan, James

VERSION 1 – REVIEW

REVIEWER	Christina Dobson Newcastle University, UK
REVIEW RETURNED	27-Mar-2020

GENERAL COMMENTS	The authors present their protocol for a novel study investigating whether purchase patterns of pain killers and indigestion tablets is indicative of early stage ovarian cancer. Overall the protocol is well written and clear. I have just a few points that I feel require further clarification: P4, lines 4-5 - you state that 'not wanting to waste the doctor's time', normalisation of symptoms and failure to make the link between their symptomatic experiences and cancer as the main reasons that people do not present to primary care. There is a large body of literature which shows that there are numerous social and systemic factors that act as barriers to presentation, for instance caring/work responsibilities, or difficulties in accessing primary care/health services, and it is important to acknowledge that delayed presentation is also often the result of factors beyond the patient's immediate control. P6, line 60 - you mention that your recruitment strategy includes dissemination of study flyers and leaflets - please can you provide more information of exactly how these flyers will be disseminated and to whom? If it is aimed to recruit individuals less likely to engage with social media how will you be defining and targeting this group? P7 - lines 29-33. Whilst I appreciate that you do not wish to name the specific high street retailers in the study I think it is important to provide a bit more context about the types of retailers - is this supermarkets or high street pharmacies, such as boots/superdrug? People are likely to purchase pain killers and indigestion medication from both and so it would be helpful to know which type of retailer's purchase data you are focusing on and why. P9, line 35 - please can you provide further information about how long electronically held data on clinical characteristics of those women with ovarian cancer will be held for
---

REVIEWER	Clare Aitken Erasmus MC University Medical Centre, Department of Public Health
REVIEW RETURNED	07-Apr-2020

GENERAL COMMENTS	Thank you for the opportunity to review this well-written and very clear study protocol. The authors have done a good job of clearly stating the aims and hypothesis of this study. There are a few opportunities to further clarify aspects of the study. MAJOR COMMENTS:  - It is a bit vague what a 'high street' retailer is for those who are not from the UK. I know that it is not possible to name the retailers that this stages, but can you give concrete examples of what kind of stores this will participate? I imagine this will include pharmacies, but you include changes in eating habits in Figure 1, so have you also included supermarkets? Also, some women may not seek help at 'traditional' pharmacies; have you included retailers that sell alternative medicines and vitamin supplements? - The hypothesis of this study is that a significant change in purchasing behaviour, primarily an increase in self-medication or self-care behaviours to treat symptoms, will be observed in the cases compared with the control group. However, I don't think 'purchasing behaviour' has been defined clearly enough in the manuscript. It seems that, based on your proof-of-concept study, there are underlying hypotheses about what kinds of purchases will change/increase (e.g. pain-relief medication, medication for bloating). Could you please clarify the definitions for purchases of self-medication and self-care behaviours? - I think the secondary aim of this study raises some ethical questions and potential dilemmas. The secondary aim is to define a 'purchase threshold' as an 'alert' for potential ovarian cancer cases. This aim sounds like the authors would like to create a population screening threshold for use in the future. However, population screening should be conducted only when in the following circumstances: a) women can make an informed choice about whether or not to participate, b) that the benefits and harms of screening are in balance and c) that all women in the target population are able to participate if they wish to. My concern is that by defining a threshold (of which sensitivity and specificity may not be adequately assessed), some women could be 'screened' without their knowledge or consent and not all women in the population will benefit from such a system (i.e. those without loyalty cards would not be included). How will the authors deal with these issues within their study? - Selection bias is a possible risk in this study, but is not defined in the sources of bias (pg. 11). The major source of selection bias is the fact that participation is limited to those with loyalty cards at the two retailers. Do the authors know if women with a loyalty card are representative of their target population? If not, how will this selection bias impact the expected results? MINOR COMMENTS:  - Introduction, pg. 3, ln. 54-55: You state "While in some cases patients may present very quickly, for some patients it could take up to 12 months before they seek any medical help." – do you have evident to support this time period? - Introduction, pg. 4, ln. 28-30: There are a few typos in this sentence: "The study suggested that there may be an increase in the purchases of pain killers and indigestion tablets up to 10-12 months before diagnosis" - Methods, Statistical Methods: Why was 1:1 matching selected?
--

REVIEWER	Aditi BHATT Zydus hospital, India
REVIEW RETURNED	13-Apr-2020

GENERAL COMMENTS	This is a case-control study that aims to study the pre-therapy purchase behaviour of ovarian cancer patients in comparison to healthy controls The manuscript is not well structured with too much focus on data collection and management. The STROBE statement is missing 1. It may be presumed that the study will include only patients with epithelial ovarian cancer? The authors have stated all subtypes of ovarian cancer.. please specify 2. Do the authors intend to look at early or advanced ovarian cancer? Please mention what FIGO stages you wish to study. Most high grade serous cancer present with advanced disease and screening has failed to diagnose these tumors early. This is largely because of the rapid evolution of these tumors into advanced disease. It is hard to believe that pretherapy purchase behaviour could make a difference and help identify these patients. Even if it is presumed it would, how would the current study affect the management of these patients? On the other hand, if the authors are targetting early stage disease, they proportion of these patients will be less and it should be specified whether the sample size is adequate to address these subgroups. 3. The inclusion and exclusion criteria should be listed properly. The authors state that no patients will be excluded- will patients who have been diagnosed in the last two years and died of disease be included? 4. The risk factors that are going to be studied should be predefined and listed. All the risk should be mentioned with an appropriate explanation and referencing for the same. 5. How do the authors propose to address comorbidities? The difference in purchasing could be due to other illnesses. How will that be addressed in the study? 6. The statistical methods are not clear. How will the cases and controls be compared is not clear. Again the statistical methods have to be properly listed- there cannot be 'or' between two kinds of analysis the authors intend to perform. 7. In risks and biases, there is no mention of the biases 8. The strengths and weakness of the study should be described in a separate section.
---

REVIEWER	Jennifer Ose Huntsman Cancer Institute, University of Utah
REVIEW RETURNED	14-Apr-2020

GENERAL COMMENTS	Brewer et al. submitted a study protocol for an observational case-control study focusing on the patient interval in ovarian cancer diagnosis using data from patients who use a loyalty card and retrospectively comparing purchase patterns. The study is unique and is using an innovative study design to use publically available data to reduce delays in ovarian cancer diagnosis. The article is relevant and well written. However before publication some minor changes need to be addressed. Please explain “high street retailer”. Please explain the variables from the clinical questionnaires in more detail and provide information on the different categories. Please be specific – will data be deleted 10 years after end of recruitment or 10 years after completion. Provide more details on the criteria for matching of cases and controls. Although the study recruitment already started, it may be worthwhile considering only selectio of controls that did NOT have a hysterectomy. Please explain how you will account for potential changes in packaging, dosing etc. in statistical analyses? If possible, at this point, consider specific evaluation of diabetes mellitus (DM) medications such as metformin. There is accumulating evidence on the role of DM in ovarian cancer. Please explain target and non-target purchases. Since ovarian cancer risk factors differ by age and menopausal status it may be useful to perform stratified analyses by these factors. Please clarify if statistical analyses will be performed for different medications separately or combined? It is recommended to adjust by number of people living in the respective households to account for the purchase of medication for others.
--

Reviewer: 1

Reviewer Name

Christina Dobson

Institution and Country

Newcastle University, UK

Please state any competing interests or state 'None declared':
None declared

Please leave your comments for the authors below

The authors present their protocol for a novel study investigating whether purchase patterns of pain killers and indigestion tablets is indicative of early stage ovarian cancer. Overall the protocol is well written and clear. I have just a few points that I feel require further clarification:

P4, lines 4-5 - you state that 'not wanting to waste the doctor's time', normalisation of symptoms and failure to make the link between their symptomatic experiences and cancer as the main reasons that people do not present to primary care. There is a large body of literature which shows that there are numerous social and systemic factors that act as barriers to presentation, for instance caring/work responsibilities, or difficulties in accessing primary care/health services, and it is important to acknowledge that delayed presentation is also often the result of factors beyond the patient's immediate control.

We agree that there are other reasons for delayed presentation and we did not include all mentioned above. Based on the help-seeking literature for ovarian cancer, we presented those that are highly relevant to our study. We now added a sentence highlighting other potential factors as well.

P6, line 60 - you mention that your recruitment strategy includes dissemination of study flyers and leaflets - please can you provide more information of exactly how these flyers will be disseminated and to whom? If it is aimed to recruit individuals less likely to engage with social media how will you be defining and targeting this group?

We have added that flyers and leaflets will be distributed to patients in participating NHS recruitment clinics.

P7 - lines 29-33. Whilst I appreciate that you do not wish to name the specific high street retailers in the study I think it is important to provide a bit more context about the types of retailers - is this supermarkets or high street pharmacies, such as boots/superdrug? People are likely to purchase pain killers and indigestion medication from both and so it would be helpful to know which type of retailer's purchase data you are focusing on and why.

We have now included further details about the definition of the retailers included in our study. We also highlighted that no pharmacy data will be recorded as that will require accessing individual health records.

P9, line 35 - please can you provide further information about how long electronically held data on clinical characteristics of those women with ovarian cancer will be held for

We have now included further details about this on the data deletion section. We hope it is clear.

Reviewer: 2

Reviewer Name

Clare Aitken

Institution and Country

Erasmus MC University Medical Centre, Department of Public Health

Please state any competing interests or state 'None declared':
None declared

Please leave your comments for the authors below Thank you for the opportunity to review this well-written and very clear study protocol. The authors have done a good job of clearly stating the aims and hypothesis of this study. There are a few opportunities to further clarify aspects of the study.

MAJOR COMMENTS:

- It is a bit vague what a 'high street' retailer is for those who are not from the UK. I know that it is not possible to name the retailers that this stages, but can you give concrete examples of what kind of stores this will participate? I imagine this will include pharmacies, but you include changes in eating habits in Figure 1, so have you also included supermarkets? Also, some women may not seek help at 'traditional' pharmacies; have you included retailers that sell alternative medicines and vitamin supplements?

A description of high street retailers is now included as health, beauty, and/or grocery retailers accessible to all UK population within a postcode sector/area. Further details about the retailers that is supporting this study is included in the methods section. We have not included retailers that sell alternative medicines, but there are many alternative medicines and vitamin supplements sold at the participating retailers.

- The hypothesis of this study is that a significant change in purchasing behaviour, primarily an increase in self-medication or self-care behaviours to treat symptoms, will be observed in the cases compared with the control group. However, I don't think 'purchasing behaviour' has been defined clearly enough in the manuscript. It seems that, based on your proof-of-concept study, there are underlying hypotheses about what kinds of purchases will change/increase (e.g. pain-relief medication, medication for bloating). Could you please clarify the definitions for purchases of self-medication and self-care behaviours?

Thank you for the comment. While we had a hypothesis in the proof of concept study to focus on specific purchases, for this study due to the number of participants and the range of data to go back up to 6 years from the date of consent, we decided not to limit the analyses to pain relief and indigestion only. While the primary focus will be on increase of purchases of self-medication, secondary analyses will explore a change in frequency or quantity of other regular purchases. We have now added more clarity on this in the manuscript on under Study aims.

- I think the secondary aim of this study raises some ethical questions and potential dilemmas. The secondary aim is to define a 'purchase threshold' as an 'alert' for potential ovarian cancer cases. This aim sounds like the authors would like to create a population screening threshold for use in the future. However, population screening should be conducted only when in the following circumstances: a) women can make an informed choice about whether or not to participate, b) that the benefits and harms of screening are in balance and c) that all women in the target population are able to participate if they wish to. My concern is that by defining a threshold (of which sensitivity and specificity may not be adequately assessed), some women could be 'screened' without their knowledge or consent and not all women in the population will benefit from such a system (i.e. those without loyalty cards would not be included). How will the authors deal with these issues within their study?

Thank you for this comment. Ultimately, in the future beyond this project, if purchase behaviours are informative that someone might be experiencing ovarian cancer symptoms or other serious disease, it would be important to use this information like suggested in an ethical way without raising concerns and with explicit informed consent. At this stage, we are not proposing a screening programme or to use this as a diagnostic tool as alone it will be insufficient. However, as part of the larger project we are conducting behavioural science studies to investigate if and how individuals would prefer to be informed about significant changes in their health behaviours informed by their commercial data or

whether this data will be more beneficial to be used for public health interventions aiming to raise awareness rather than individual interventions. We have now added a sentence to clarify how this aim will be important for future research.

- Selection bias is a possible risk in this study, but is not defined in the sources of bias (pg. 11). The major source of selection bias is the fact that participation is limited to those with loyalty cards at the two retailers. Do the authors know if women with a loyalty card are representative of their target population? If not, how will this selection bias impact the expected results?

Thank you for pointing this out. Approximately 73% of women aged 50-80 years in the UK have a loyalty card to one of the participating high street retailers, and therefore the cardholders in CLOCS should be representative of the target population. However, we acknowledge there will still be selection bias against those who do not hold a loyalty card. We have now addressed this in the Study risks and sources of bias section of the manuscript.

MINOR COMMENTS:

- Introduction, pg. 3, ln. 54-55: You state "While in some cases patients may present very quickly, for some patients it could take up to 12 months before they seek any medical help." – do you have evidence to support this time period?

A reference for this is now included in the manuscript by Hamilton et al 2009.

- Introduction, pg. 4, ln. 28-30: There are a few typos in this sentence: "The study suggested that there may be an increase in the purchases of pain killers and indigestion tablets up to 10-12 months before diagnosis"

Thank you for pointing out this typo, we have fixed it.

- Methods, Statistical Methods: Why was 1:1 matching selected?

We chose 1:1 matching due to the expectation that recruiting healthy participants might prove harder than ovarian cancer cases, and the statistical power calculations suggested this would have sufficient power for this analysis. Also there would be extra costs involved in planning for more controls. However, we did discuss using 2:1 ratios of control:case and it is in fact in the risk register for the grant proposal as a contingency plan should we require further statistical power.

Reviewer: 3

Reviewer Name

Aditi BHATT

Institution and Country

Zydus hospital, India

Please state any competing interests or state 'None declared':
None declared

Please leave your comments for the authors below This is a case-control study that aims to study the pre-therapy purchase behaviour of ovarian cancer patients in comparison to healthy controls

The manuscript is not well structured with too much focus on data collection and management. We thank the reviewer for highlighting what we believe is a very important aspect of this project. We aimed to structure the protocol in a way that will facilitate replication/adoption by other researchers interested in using commercial data as we are reporting the first study to our knowledge to use commercial data from retailers with individual consent for a health related research. There is an increase in the interest to investigate commercial data but the pathways to do this ethically have not yet been reported. We also believe that this project needs to be completely transparent to the

researchers and to the public to ensure that all the information available to others is consistent in all sources of dissemination. We hope these reasons will be satisfactory for why we are not changing the structure of the manuscript.

The STROBE statement is missing

We will attach the STROBE checklist we intend to use for reporting results as supplementary material, and this is referred to under Ethics and dissemination.

1. It may be presumed that the study will include only patients with epithelial ovarian cancer? The authors have stated all subtypes of ovarian cancer.. please specify

The clinical questionnaire collects information about which type of ovarian cancer patients were diagnosed with, including high-grade serous, low-grade serous, endometrioid, clear cell, mucinous, borderline, and other subtypes. We have now specified this list of ovarian cancer subtypes in the inclusion criteria.

2. Do the authors intend to look at early or advanced ovarian cancer? Please mention what FIGO stages you wish to study. Most high grade serous cancer present with advanced disease and screening has failed to diagnose these tumors early. This is largely because of the rapid evolution of these tumors into advanced disease. It is hard to believe that pretherapy purchase behaviour could make a difference and help identify these patients. Even if it is presumed it would, how would the current study affect the management of these patients?

On the other hand, if the authors are targetting early stage disease, they proportion of these patients will be less and it should be specified whether the sample size is adequate to address these subgroups.

Ovarian cancer patients diagnosed at any stage are eligible. The number of participants diagnosed with each stage will be the deciding factor on whether or not we have the power to look at early or advanced ovarian cancer. The sample sizes were not designed to have sufficient statistical power to investigate subgroups, although we have pre-planned subgroup analyses to stratify on risk profiles, stage and histology.

3. The inclusion and exclusion criteria should be listed properly. The authors state that no patients will be excluded- will patients who have been diagnosed in the last two years and died of disease be included?

Eligible ovarian cancer patients will be alive at time of recruitment, and we have added women diagnosed 'and living' with ovarian cancer to the inclusion criteria.

4. The risk factors that are going to be studied should be predefined and listed. All the risk should be mentioned with an appropriate explanation and referencing for the same.

The risk factors that we will study are predefined and listed in the description of the risk factor questionnaire. We have now listed them in the statistical methods as well.

5. How do the authors propose to address comorbidities? The difference in purchasing could be due to other illnesses. How will that be addressed in the study?

We are not currently including information on comorbidities and there are two reasons for this. First, we would assume that people with comorbidities will already be receiving prescribed medication which should not be in the over-the counter purchase data we will receive. Common comorbidities are expected to be present in similar frequencies among patients with ovarian cancer and non-ovarian cancer participants. The risk factor questionnaire will address some of these comorbidities, and we will then be able to adjust for these in the analysis. Secondly, if someone is buying medication or other products for other illnesses and chronic conditions these would be observed as consistent behaviours whereas a change associated with ovarian cancer symptoms may be more nuanced in frequency depending on symptoms patients have reported. We have included questions on symptoms associated with women's menstruation cycle and menopause to check/adjust for chronic purchasing behaviours for those symptoms among the healthy cohort. We have addressed this in the Study risks and sources of bias section.

6. The statistical methods are not clear. How will the cases and controls be compared is not clear. Again the statistical methods have to be properly listed- there cannot be 'or' between two kinds of analysis the authors intend to perform.

We have now fixed the 'or' typo in the statistical methods so there is no confusion about the kinds of analysis we intend to perform. We have added more details about analysing different medications separately at first, and then in combinations. The primary focus for analyses will be on increase of purchases of self-medication, and secondary analyses will explore a change in frequency or quantity of other regular purchases. We have now added more clarity on this in the manuscript on under Study aims.

7. In risks and biases, there is no mention of the biases

In the Study risks and sources of bias section, we discuss recall bias and now selection bias also.

8. The strengths and weakness of the study should be described in a separate section.

The strengths and weaknesses of the study are described separately at the beginning of the manuscript.

Reviewer: 4

Reviewer Name

Jennifer Ose

Institution and Country

Huntsman Cancer Institute, University of Utah

Please state any competing interests or state 'None declared':
None declared

Please leave your comments for the authors below Brewer et al. submitted a study protocol for an observational case-control study focusing on the patient interval in ovarian cancer diagnosis using data from patients who use a loyalty card and retrospectively comparing purchase patterns. The study is unique and is using an innovative study design to use publically available data to reduce delays in ovarian cancer diagnosis. The article is relevant and well written. However before publication some minor changes need to be addressed.

Please explain "high street retailer".

A description of high street retailers is now included as health, beauty, and/or grocery retailers accessible to all UK population within a postcode sector/area. Further details about the retailers that is supporting this study is included in the methods section

Please explain the variables from the clinical questionnaires in more detail and provide information on the different categories.

We have now added a list of variables and the different categories of these variables in the clinical questionnaire.

Please be specific – will data be deleted 10 years after end of recruitment or 10 years after completion.

We have addressed this in the Data deletion session and explained what type of data will be deleted and when.

Provide more details on the criteria for matching of cases and controls.

We will be matching cases and controls on age and ethnicity. Age will be matched in 5-year age groups, and ethnicity will be matched based on guidance from the Office for National Statistics for recommended ethnic groups. We have added this detail and reference to the manuscript.

Although the study recruitment already started, it may be worthwhile considering only selection of controls that did NOT have a hysterectomy.

Thank you for this suggestion. Our risk factor questionnaire collects information about whether or not controls have had a hysterectomy, allowing us to restrict analyses to compare cases and controls who have not had a hysterectomy.

Please explain how you will account for potential changes in packaging, dosing etc. in statistical analyses?

Thank you for your comment. The high street retailers have provided us with the hierarchy of products and product descriptions, allowing us to finely categorise by packaging and dosing. We will also address this in the code, e.g. compute the dosage from different purchases and then categorize painkillers purchases based on dose per month.

If possible, at this point, consider specific evaluation of diabetes mellitus (DM) medications such as metformin. There is accumulating evidence on the role of DM in ovarian cancer.

We agree this would be interesting to investigate, however loyalty card data only collects information on over-the-counter purchases, and we will not have access to prescription medication data.

Please explain target and non-target purchases.

Target purchases are items that are suspected to treat ovarian cancer symptoms such as indigestion tablets, and non-target purchases refer to all other items. We have now added descriptions of target and non-target purchases to the manuscript.

Since ovarian cancer risk factors differ by age and menopausal status it may be useful to perform stratified analyses by these factors.

We agree with this comment. Cases and controls will be matched in 5-year age groups in the matched analysis, and we will be able to adjust for menopausal status in the conditional logistic regression model.

Please clarify if statistical analyses will be performed for different medications separately or combined?

We have now clarified that we will perform analyses for different medications separately and combined in the statistical methods section.

It is recommended to adjust by number of people living in the respective households to account for the purchase of medication for others.

We agree that this would help account for the purchase of medication for individuals other than the CLOCS participant and are amending the questionnaire to add this question.

VERSION 2 – REVIEW

REVIEWER	Clare Aitken Erasmus MC University Medical Center Rotterdam, Department of Public Health
-----------------	--

REVIEW RETURNED	15-May-2020
-------------

GENERAL COMMENTS	The authors have done a good job addressing the comments from the reviewers. There are a few further opportunities to refine the manuscript. Comments:  • I think that the line added to the 'Methods and Analysis' section of the abstract (“... (i.e. all shops accessible within a postcode sector)”) is less clear than what was in the previous draft. I would consider changing this back to the original. • I think UK readers will be able to make assumptions about which high store retailers the authors are collaborating with. I am still unsure how far the authors are really able to measure 'eating habits' based on loyalty card data from such retailers. If my assumptions about the retailers are correct, these are not stores in which people generally shop for groceries. Therefore, I suspect it is not a clear or complete representation of eating habits. I suggest clarifying this in the methods, and possibly adding this as a limitation. • I still think that the target and non-target purchases are not defined clearly enough. These endpoints should be more clearly defined, as I have the (perhaps incorrect) impression that the analysis is somewhat exploratory. I suggest adding a list of target purchases (which painkillers? which indigestion medication?) to the supplementary materials. • The authors state that “many women with comorbidities are likely already receiving prescribed medication to treat these”. Can you provide evidence for this statement?
---

REVIEWER	Aditi BHATT Zydus Hospital, India
REVIEW RETURNED	22-May-2020

GENERAL COMMENTS	The authors have revised the manuscript but the concerns are still not adequately addressed.  1. There are some basic errors like not correctly defining the histological subtypes- its not clear whether non-epithelial tumors are included or excluded- germ cell tumors, sex cord stromal tumors. One term ' epithelial ovarian cancer' that should be mentioned is missing 2. Whilst it is important to focus on the method of data collection so that the study is reproducible, the more important concern would be the clinical impact- on how this study is going to help in diagnosing ovarian cancer early which the authors are not able to bring out in the manuscript. 3. Not matching cases and controls according to comorbidities is an important limitation. Though these patients receive prescribed medications, use of the target purchases can still be related to the comorbidity- for e.g. exacerbation of arthritis leading to an increase in purchase of analgesics. 4. The authors have described the statistical methods but there is no description of the risk models/scores that will be used and how risk stratification will be performed and applied. Citing the reference alone is not enough. What proportion of patients are expected to be at high risk in the study and is the study powered to establish a difference between the high and low risk groups. 5. Sections on data management and public involvement can be provided as supplements with only a brief discussion in the main
--

	manuscript. 6. The authors should discuss the potential clinical impact of the results on future management/early diagnosis of ovarian cancer. Though the goals have been listed, it is still unclear how the results of this study can be applied to clinical practice
REVIEWER	Jennifer Ose Huntsman Cancer Institute, University of Utah
REVIEW RETURNED	20-May-2020
GENERAL COMMENTS	The authors have addressed all my comments. I don't have any additional comments.

VERSION 2 – AUTHOR RESPONSE

Reviewer: 2

Reviewer Name

Clare Aitken

Institution and Country

Erasmus MC University Medical Center Rotterdam, Department of Public Health

Please state any competing interests or state 'None declared': None declared

Please leave your comments for the authors below The authors have done a good job addressing the comments from the reviewers. There are a few further opportunities to refine the manuscript.

Comments:

- I think that the line added to the 'Methods and Analysis' section of the abstract (“... (i.e. all shops accessible within a postcode sector”) is less clear than what was in the previous draft. I would consider changing this back to the original.

We have now removed this and reverted to the original version.

- I think UK readers will be able to make assumptions about which high street retailers the authors are collaborating with. I am still unsure how far the authors are really able to measure 'eating habits' based on loyalty card data from such retailers. If my assumptions about the retailers are correct, these are not stores in which people generally shop for groceries. Therefore, I suspect it is not a clear or complete representation of eating habits. I suggest clarifying this in the methods, and possibly adding this as a limitation.

Thank you for your comment. Firstly, the assumption the reviewer made is incorrect. One of the participating retailers is a grocery store which will represent eating habits. We generated and adopted the descriptions from the retailers' websites but unless it is approved by them, we cannot use their exact wording to describe them. We have now added details for further clarity to the Methods section. Based on this, we hope that our access to dietary behaviours will be further clarified.

- I still think that the target and non-target purchases are not defined clearly enough. These endpoints should be more clearly defined, as I have the (perhaps incorrect) impression that the analysis is somewhat exploratory. I suggest adding a list of target purchases (which painkillers? which indigestion medication?) to the supplementary materials.

We appreciate that target purchases needed more clarification. We have added a sentence to detail some target items such as pain, indigestion, and bloating relief medications. We have also added Table 1 to Appendix A in the supplementary file with examples of such purchases.

- The authors state that “many women with comorbidities are likely already receiving prescribed medication to treat these”. Can you provide evidence for this statement?

We have now reworded this section to provide clarification on what we aimed to describe and how we will manage the risk of bias resulting from people with comorbidities. We have also added examples of comorbidities that are associated with ovarian cancer and provided a reference in the manuscript.

Reviewer: 4

Reviewer Name

Jennifer Ose

Institution and Country

Huntsman Cancer Institute, University of Utah

Please state any competing interests or state 'None declared': No competing interests

Please leave your comments for the authors below The authors have addressed all my comments. I don't have any additional comments.

Reviewer: 3

Reviewer Name

Aditi BHATT

Institution and Country

Zydus Hospital, India

Please state any competing interests or state 'None declared': None declared

Please leave your comments for the authors below The authors have revised the manuscript but the concerns are still not adequately addressed.

1. There are some basic errors like not correctly defining the histological subtypes- its not clear whether non-epithelial tumors are included or excluded- germ cell tumors, sex cord stromal tumors.

One term 'epithelial ovarian cancer' that should be mentioned is missing

Under Inclusion/Exclusion Criteria for cases, we have included the subtypes of epithelial ovarian cancers, and we have now added the term 'epithelial ovarian cancer' for maximum clarification. We also specified that women with any type of ovarian cancer who also meet the other criteria are eligible.

2. Whilst it is important to focus on the method of data collection so that the study is reproducible, the more important concern would be the clinical impact- on how this study is going to help in diagnosing ovarian cancer early which the authors are not able to bring out in the manuscript.

The purpose of this study is to explore whether changes in purchases of items used to self-treat ovarian cancer symptoms can help detect ovarian cancer earlier and to potentially improve recognition of symptoms so that women will seek medical advice earlier. We plan to validate the results of this study using a prospective cohort study before any clinical impact will be realised. If our study is successful, we expect our results will identify a purchase threshold upon which the probability of an ovarian cancer diagnosis is higher. The clinical impact that we imagine will be one of two things: first could be an alert sent to individuals somehow to visit a GP to discuss their symptoms and secondly a potential public health message about those purchase thresholds and self-treatment of ovarian cancer symptoms. We have now added a statement to the manuscript about clinical impact. As part of our study and ongoing research, we will be conducting behavioural science experiments and focus groups to determine how best to provide this alert back to individuals.

3. Not matching cases and controls according to comorbidities is an important limitation. Though these patients receive prescribed. medications, use of the target purchases can still be related to the comorbidity- for e.g. exacerbation of arthritis leading to an increase in purchase of analgesics.

Thank you for your point. We will keep this in mind when interpreting the results of this study and will consider this as a limitation in further outputs.

4. The authors have described the statistical methods but there is no description of the risk models/scores that will be used and how risk stratification will be performed and applied. Citing the reference alone is not enough. What proportion of patients are expected to be at high risk in the study and is the study powered to establish a difference between the high and low risk groups.

We will establish the risk profile, a mathematical summation of all of the risk factors, of each participant based on the method used in the reference. Our study is underpowered to establish a difference between high and low risk groups, it is therefore exploratory. If it does show an indication of a difference, then that can guide future studies that will be sufficiently powered to address this. We have discussed this in the Sample size and statistical power section of the manuscript. We plan to

look at quintiles of risk distribution to understand the proportion of high and low risk participants. The statistical analysis plan entails looking at quintiles of the risk distribution, and therefore it will be 20% of participants in the higher risk group compared to the rest of the quintiles.

5. Sections on data management and public involvement can be provided as supplements with only a brief discussion in the main manuscript.

These sections are set out by the BMJ Open as compulsory sections to be included in the protocol, we are happy to cut this down if the Editors also feel that the information can be included elsewhere.

6. The authors should discuss the potential clinical impact of the results on future management/early diagnosis of ovarian cancer. Though the goals have been listed, it is still unclear how the results of this study can be applied to clinical practice

Please see our response to point 2.

VERSION 3 – REVIEW

REVIEWER	Clare Aitken Erasmus MC University Medical Center, the Netherlands
REVIEW RETURNED	29-Jun-2020
GENERAL COMMENTS	I think that the authors have addressed the concerns of the reviewers sufficiently and I have no further comments on the manuscript.
REVIEWER	Aditi BHATT Zydus Hospital, India
REVIEW RETURNED	22-Jun-2020
GENERAL COMMENTS	The explanation provided by the authors for points 2 and 4 raised by this reviewer in the last review should be added to the manuscript to offer greater clarity to the readers.

VERSION 3 – AUTHOR RESPONSE

Reviewer: 3

The explanation provided by the authors for points 2 and 4 raised by this reviewer in the last review should be added to the manuscript to offer greater clarity to the readers.

Point 2

The findings from this retrospective study will need to be validated in a prospective cohort study prior to any clinical impact being realised. However, the clinical impact of this study for ovarian cancer diagnosis may involve a number of potential interventions. Future interventions could include an alert sent to individuals who exceed the “alert threshold” to visit a GP to discuss their symptoms, or a potential public health message about those purchase thresholds and self-treatment of ovarian cancer symptoms.

Point 4.

We will establish the risk profile, a mathematical summation of all of the risk factors, of each participant based on the published method [17]. Our study will be underpowered to establish a difference between high and low risk groups (20% of subjects per quintile of risk), it is therefore exploratory. We will estimate the predictive values of this “early diagnosis” test for each stratum. If this analysis indicates a potential difference, then that will guide future studies that will be sufficiently powered to address this.